# Factors Impacting Retention of Aged Care Workers: A Systematic Review

**DOI:** 10.3390/healthcare11233008

**Published:** 2023-11-21

**Authors:** Claire Thwaites, Jonathan P. McKercher, Deirdre Fetherstonhaugh, Irene Blackberry, Julia F-M. Gilmartin-Thomas, Nicholas F. Taylor, Sharon L. Bourke, Sally Fowler-Davis, Susan Hammond, Meg E. Morris

**Affiliations:** 1Academic and Research Collaborative in Health, La Trobe University, Melbourne, VIC 3086, Australia; j.mckercher@latrobe.edu.au (J.P.M.); n.taylor@latrobe.edu.au (N.F.T.); s.bourke@latrobe.edu.au (S.L.B.); m.morris@latrobe.edu.au (M.E.M.); 2Victorian Rehabilitation Centre, Healthscope, Glen Waverley, Melbourne, VIC 3150, Australia; hammondss@bigpond.com; 3Australian Centre for Evidence Based Aged Care, La Trobe University, Melbourne, VIC 3086, Australia; d.fetherstonhaugh@latrobe.edu.au; 4Care Economy and Research Institute and John Richards Centre for Rural Ageing Research, La Trobe University, Albury-Wodonga, VIC 3086, Australia; i.blackberry@latrobe.edu.au; 5School of Allied Health, Human Services and Sport, La Trobe University, Melbourne, VIC 3086, Australia; j.gilmartin-thomas@latrobe.edu.au; 6Allied Health Department, Alfred Health, Melbourne, VIC 3181, Australia; 7Eastern Health, Box Hill, VIC 3128, Australia; 8School of Nursing and Midwifery, La Trobe University, Melbourne, VIC 3086, Australia; 9Anglia Ruskin University, Bishop Hall Lane, Chelmsford CM1 1SQ, UK; sally.fowler-davis@aru.ac.uk

**Keywords:** aged care, workforce, retention, ageing, nursing, allied health, care quality, migrant workers

## Abstract

Retention of care support workers in residential aged care facilities and home-based, domiciliary aged care is a global challenge, with rapid turnover, low job satisfaction, and poorly defined career pathways. A mixed-methods systematic review of the workforce literature was conducted to understand the factors that attract and retain care staff across the aged care workforce. The search yielded 49 studies. Three studies tested education and training interventions with the aim of boosting workforce retention and the remaining 46 studies explored opinions and experiences of care workers in 20 quantitative, four mixed-methods and 22 qualitative studies. A range of factors impacted retention of aged care staff. Two broad themes emerged from the analysis: individual and organisational factors facilitating retention. Individual factors related to personal satisfaction with the role, positive relationships with other staff, families, and residents, and a cooperative workplace culture. Organisational factors included opportunities for on-the-job training and career development, appropriate wages, policies to prevent workplace injuries, and job stability. Understaffing was often cited as a factor associated with turnover, together with heavy workloads, stress, and low job satisfaction. With global concerns about the safety and quality of aged care services, this study presents the data associated with best practice for retaining aged care workers.

## 1. Introduction

Globally, one of the most rapidly growing sectors of the care economy is aged care [1]. People in developed countries can expect to live at least a decade longer than the global average of 73 years [2]. Of concern, the proportion of the population requiring health and care services is increasing due to an ageing demographic [3]. At the same time, the share of the population that is able to deliver services is decreasing [4]. The size of the frontline aged care workforce, such as personal care attendants, nursing assistants, and allied health assistants, will need to quadruple to meet the demand of ageing societies [5]. The aged care workforce is reliant on a supply of workers that predominantly consists of women with few qualifications, in labour-intensive roles [6,7]. These staff provide essential management of mobility, cognitive impairment, toileting, bathing, and feeding [8]. It is estimated that more than half of aged care residents live in facilities with insufficient staffing for person-centred care [9].

Upwards of 25% of aged care workers in countries such as the United Kingdom and Australia spend less than one year in care support roles [10,11,12]. The median annual turnover for nursing assistants in the United States is nearly 99% [13]. Staff turnover in aged care settings is negatively associated with the quality of care [13,14]. Turnover is also associated with falls [15], infection rates [16], and low resident and staff satisfaction [17,18]. Supervisors of the care workforce are usually registered and enrolled nurses who also have critical staff shortages [19]. To improve care experiences and outcomes for older people, care workers need to be trained, safe, and adequately supported to be retained in the workforce.

Evidence-informed, tailored strategies for workforce redesign are arguably needed to recruit and retain aged care workers [20] and to prevent burnout [21] and safety incidents [22]. A variety of rewards are recognised and valued by staff [23]. Incentives include adequate pay rates [24], safe and positive working conditions [25,26], and suitable geographical locations for employment [27]. The extent to which other determinants of workforce retention, such as comprehensive orientation and ongoing training at the workplace, apply to aged care workers is currently not known [28].

There is a need to better align the motivations and needs of aged care workers to the requirements of aged care support roles, to improve the length of service tenure [23]. The main aim of this systematic review is to identify, summarise, and aggregate the quantitative and qualitative evidence related to the retention of care workers in aged care homes, respite care, retirement villages, and home settings. A second aim is to identify barriers and facilitators to care staff retention in the aged care sector.

## 2. Materials and Methods

A mixed-methods design following the JBI convergent integrated approach [29] was used to synthesise quantitative and qualitative evidence. This approach reflected the complexity of research questions in health and social care and afforded deep understanding of the issues [30]. The results were reported in accordance with the preferred reporting for systematic reviews and meta-analyses (PRISMA) guidelines [31] (Appendix A) and the enhancing transparency in reporting the synthesis of qualitative research (ENTREQ) checklist [32] (Appendix A). The review was prospectively registered with the International Prospective Register of Systematic Reviews (PROSPERO) (CRD:42023440055).

### 2.1. Search Strategy

A systematic literature search identified articles from Medline, Embase, PsycINFO, CINAHL, and AgeLine. Articles were limited to peer-reviewed, primary studies published in the English language from January 2012 to July 2023 to provide a contemporary overview of the evidence. No geographical limitations were placed on the search. The search was developed and conducted with a university academic librarian, with keywords informed by MeSH headings and focused on the concepts of population (i.e., nursing assistant and other relevant terms), setting (i.e., residential aged care), and outcomes of interest. These outcomes included, but were not restricted to (1) workplace satisfaction: job status, turnover or intention to leave, stress (burnout, occupation, job), attitudes, perceptions, workplace violence or danger, absenteeism, job fulfilment, conflict resolution; (2) barriers and facilitators: motivation, supports, challenges; (3) workload: rostering, supervision, understaffing, recruitment, retention, empathy or compassion fatigue; (4) professional development: education, ongoing training, job or career flexibility, professional identity, judgement, career opportunities and pathways; (5) clinical environment: working conditions, leadership, manager support, mentorship, occupational health and safety; and (6) interventions to optimise recruitment and retention. Additional articles were searched for via reference lists of included studies, relevant systematic reviews, trial registries (i.e., WHO International Clinical Trials Registry Platform, ClinicalTrials.gov, and Australian New Zealand Clinical Trials Registry), and grey literature (i.e., Google Scholar, social and aged welfare organisational websites). The search strategy for Medline is detailed in Appendix A. Identified articles were imported into Covidence [33] and duplicates removed.

### 2.2. Eligibility and Screening

An aged care worker was defined as a person employed to independently deliver direct care under the overall supervision of a registered or enrolled nurse [34] or under the supervision of an allied health professional or medical practitioner, within an aged care setting (residential aged care or home-based aged care). A widely accepted definition of older person is over 60–65 years [35]; however, we did not apply age limits to inclusion criteria. Residential aged care facilities (also known as care homes, long-term care facilities or nursing homes) were defined as facilities providing 24 h nursing and care to older adults [36]. Home-based care (domiciliary care) was defined as the provision of scheduled, as-needed, healthcare within an older person’s home. In the case of a study including both aged care workers and other employees within aged care, the study was only included if the data related to aged care workers could be separated or the number of aged care support workers represented a clear majority (>70%) of participants. Analysis of the professional care workforce such as doctors, nurses, and physiotherapists was not performed as this has been covered in a separate manuscript [37]. Datasets were limited to the past 10 years to focus on contemporary evidence concerning the aged care sector, noting changes including the removal of distinction between high- and low-level care, establishment of “home care” and means testing for contributions to care [38]. The inclusion and exclusion criteria are listed in Table 1. The title, abstract and full-text articles were screened independently by two authors (C.T., J.M.) who applied the criteria from Table 1 to determine eligibility. Disagreements were discussed and resolved, involving a third reviewer (M.M.) when necessary.

### 2.3. Methodological Quality

The mixed-methods appraisal tool (MMAT) [39] was used by two authors independently (C.T., J.M.) to assess methodological quality of the selected studies. Consensus was achieved through discussion among the two reviewers, with incorporation of a third author (M.M.) as required. The design of the MMAT allows for analysis of studies across a range of methods, with different criteria to describe and evaluate mixed methods, qualitative and quantitative studies. These criteria assess factors that impact risk of bias, completeness, and transparency of studies. Following updated recommendations [40], MMAT scores were not allocated a numerical value; instead, studies were ranked low, moderate, or high quality. Due to the limited literature, all studies were included in data synthesis; however, study quality was considered in interpreting the synthesised evidence.

### 2.4. Data Extraction and Analysis

A data extraction sheet was purposefully developed, based on the JBI mixed methods data extraction form [30], and used to collect data related to the study aims, setting, design and methodological framework, sampling methods, data collection methods, demographics of the care worker populations, and key findings related to review outcomes. For interventional studies, an abbreviated template for intervention description and replication TIDieR checklist [41] was used to capture details of the intervention, who provided it, and how it was delivered. A single author (C.T.) completed this process with data checked for accuracy, completeness, and quality by another author (M.M.). Where studies included aged care support workers and other employees in aged care, only the data relating to care workers were extracted where possible.

Quantitative data were tabulated, with descriptive statistics, *p*-values, and effect sizes for mean score differences presented where available. The results were then narratively synthesised. Originally, the research team had planned to calculate change scores where relevant data, such as workforce attrition or retention rates, were sufficiently reported; however, limited data availability prevented this. Qualitative data pertaining to the research question were extracted and managed using NVivo software (release 1.7) [42]. An interpretive description approach was followed to allow for representation of subjective experiences at both individual and wider population levels [43]. The data were analysed thematically, with initial reading and rereading for deep understanding [44]. Key words, phrases, and sentences relevant to the research question were synthesised and codes developed. These were refined and grouped into themes and subthemes, then reviewed and discussed (C.T., M.M.) until consensus reached.

Following the JBI mixed methods review methodology [30], the results of quantitative and qualitative components were aggregated. The narratively synthesised quantitative results were categorised and summarised to create subcategories to fit together with the qualitative synthesis. This generated a new set of organised findings regarding factors influencing aged care worker retention. A subgroup analysis pertaining to the care setting was not performed because there were minimal home-based care studies meeting the inclusion criteria.

## 3. Results

### 3.1. Identified Studies

As seen in the PRISMA flow diagram (Figure 1) [31], a total of 2506 studies were identified through database searches, with an additional 29 studies found through citation searching, websites, or organisation releases. Following removal of duplicates, 2460 studies were independently screened for eligibility, with 143 assessed as potentially eligible. Some studies (*n* = 26) were excluded because data related to aged care workers could not be separated from professionals or administrators working in aged care. Some were excluded because data were derived from assessments that occurred prior to 2012 (*n* = 21), or outcomes were not related to factors influencing retention of aged care workers (*n* = 16). In total, 49 peer-reviewed, English language studies reporting primary research results were included (23 quantitative studies, 4 mixed-methods investigations, and 22 qualitative analyses). Grey literature was reviewed by two assessors. This led to identifying gaps in research needing to be addressed, including exploration of the specific needs of migrant care workers, the workforce challenges in respite aged care, and strategies to improve recruitment of the aged care workforce, particularly in regional and remote communities.

### 3.2. Study Characteristics

The characteristics of studies are presented in Table 2, Table 3 and Table 4. Three studies implemented strategies to improve retention. Of the studies applying an intervention, two were randomised controlled trials [45,46] and one used participatory action research with pre–post assessments [47]. The use of descriptive surveys and questionnaires with a cross-sectional (*n* = 20) or longitudinal (*n* = 2) design were reported in all quantitative descriptive and mixed-methods studies. The exception was the trial by Sharma et al. (2022) [48] who used human resource records to assess the relationship between wages and aged care staff turnover. Qualitative study designs included interpretative (*n* = 18), long ethnographic (*n* = 2), grounded theory (*n* = 1), and phenomenological approaches (*n* = 1). For these studies, individual interviews were most common (*n* = 14), or focus groups (*n* = 5), or a combination of data collection methods (*n* = 3). Appendix A show that purposive sampling was most favoured (*n* = 33). Appendix A highlight a wide range of outcomes relevant to care worker retention. These were captured using surveys on workplace culture, work-related injuries, job satisfaction, leadership styles, mental health of staff, intention to leave, wages, work-related stress, and the impact of the COVID-19 pandemic on care workers.

Studies were conducted across the globe, with most in the United States (*n* = 18), followed by Australia (*n* = 10), Canada (*n* = 5), Sweden (*n* = 5), and Taiwan (*n* = 4). Others were from Hong Kong, China, Korea, Austria, France, Denmark, the Netherlands, and Slovenia, providing global insight into factors influencing aged care worker retention. Sample sizes varied. For example, they ranged from the Charlesworth et al. (2020) [53] subgroup analysis of a nationwide aged care workforce survey, investigating the effect of migrant status on casual and underemployment amongst 7114 personal care attendants from residential and home aged care services, to seven migrant aged care workers providing insight on their experiences working in regional areas in the study by Winarnita et al. (2022) [93]. Appendix A show that 77–100% participants were female. Factors related to migrant aged care workers were explored in nine studies [53,56,59,64,66,72,81,88,93]. Overall, the studies included people employed as personal care attendants, certified and noncertified nurse assistants, direct care workers, patient support workers, home health assistants, and allied health assistants in addition to professional and administrative staff. Most studies (n = 47) were conducted in residential aged care facilities. Only four studies included staff working in home-based care [53,80,81,86].

### 3.3. Quality Assessment

Forty-four studies achieved a rating of “high” on the MMAT (Table 2, Table 3 and Table 4). The remaining five studies were evaluated as “moderate quality” [46,60,70,77,81]. The main reasons for lower scores related to a lack of detail regarding the statistical analysis [70,77], measurement tools or methodological choices [60,81], the numbers of non-responders [60,81], the management of confounders [46], and whether an intervention was delivered as intended [46].

### 3.4. Convergent Qualitative Synthesis

Data synthesis was aligned to two main themes of individual and organisational factors pertaining to staff retention. Individual or personal factors were defined as variables important to the care workers because of personal beliefs and life experience, in or out of the workplace. Organisational factors were defined as variables that could be changed or influenced by the organisation with impact on the care workers. These were derived from thematic analysis of the included studies, with the ecological framework of McLeroy (1988) [94] used as a stimulus during the initial formation of themes to conceptualise and understand aged care worker retention. According to the McLeroy framework, levels of influence are factors that examine relationships of individuals within their workplaces, communities, and broader society. Detailed synthesised findings are found in Appendix A.

### 3.5. Individual Influences on Staff Retention

The main factors that workers thought made them want to stay working in aged care included personal satisfaction and positive feelings of identity with their work roles [51,62,65,72,73,75,77,81,82,83,84,85,86,89,92,93]; support for their emotional and mental health [67,73,74,79]; systems to prevent work-related injuries [54,66,70]; validation, support, and positive feedback from managers and aged care organisations [62,65,82,83,88,92]; and the presence of peers from a similar cultural background [81,88,93]. Retention of older workers was positively influenced by flexibility of casual employment [91] and optimal utilisation of care worker skills [52]. Retention was better when staff had affirmative feelings towards being able to provide compassionate and nurturing care [72,73,82,84].

Positive relationships were key. The relationships between care workers and peers [75,81,82,88,91,92,93], supervisors [45,60,74,81,93], care home management [49,58,79,81,88], residents [73,75], and with residents’ families [63,84,87] were explored by many studies in this review, with consideration of workplace culture [69,73,75,76,77,79,82,83,84,90,91,92], hierarchy of care roles [49,58,73,75,76,82,84,86,87,90], and potential effect on care worker retention. Socially supportive colleagues [91,92]; well-functioning teams with open communication, collaboration, and cooperation [68,73,81,83,87,88]; and access to responsive leadership [79,93] with low management turnover [49,58] all contributed to positive workplace relationships and retention. Elements of caregiving such as forming strong attachments, experiencing genuine affection for residents [73,75,93], and concern for resident wellbeing [72,75] were also factors contributing to care workers remaining in their workplaces.

Three intervention studies sought to improve individual factors for aged care workers (Appendix A). The randomised trial by O’Brien et al. (2019) [47] reported that group-based cognitive behavioural training promoting flexibility in response to workplace stress in care home staff significantly reduced staff absences and mental health symptoms. The participatory action research by Ericson-Lindman et al. (2017) [46] involved collaborative, group-based training with registered nurses and nursing assistants together, whereby scenarios involving troubled conscience in the workplace were explored. By learning how to constructively deal with feelings of being unable to deliver a quality of care expected of themselves, participants noted an improvement in work-related performance and social support between workers. The trial by Jeon et al. (2015) [45] applied a randomised design to assess the effect of providing care home middle managers with a 12-month supported leadership program and found that both supervisor support and management behaviour towards care workers significantly improved; however, turnover and intention to leave did not differ between groups.

Factors making people want to leave included feeling a lack of emotional support following the death of elderly clients [78,80,86], perceived stigmatisation and lack of respect associated with the care role [73,75,84,85,86,89], experiences of discrimination [72,88], language barriers [72,81,88,93], lack of cultural sensitivity training, poor uptake of diversity education by managers and peers [81,88], and workplaces with perceived bullying or mistrust between care workers [75,82,88,93]. Some participants also expressed feelings of disempowerment due to limited influence on care decisions [49,58,75,86,87] or disrespect from managers [73,76,82,84,90] as being key contributors to turnover. Difficulty navigating relationships with residents and their families was sometimes reported in aged care, as well as threatening behaviours, bullying, or emotional abuse by families [63,72].

### 3.6. Organisational Factors

At a macro-organisational level, local rules, regulations, opportunities, and constraints affected staff retention. There were some reports of a lack of continuous, competent staff [74,91] and perceived understaffing [77,78,79,81,84,89,90,91,92]. This was thought to limit opportunities for person-centred care [76,85,86,92] and increased turnover [85]. Whilst casual employment offered flexibility [56,72,93] and higher payrates per hour, care workers expressed a desire for schedule control [77,89]. Teams where care workers were empowered to manage rostering found absences less impactful [50]. Having job stability was found to be important in reducing workplace stress [91] and increased retention [69]. However, many care workers reported lower job stability when providing home care, compared to those employed by a residential care facility, due to clients’ ability to enact personal preferences for care workers [86].

Local rules and role structures adversely affected care workers, with burnout [71] and low job satisfaction associated with long shift lengths, split shifts, or insufficient break allowances [71,73,86]. Heavy workloads [69], uncertainty of scope of practice [86], and limited capacity for older workers to perform nonphysical tasks [91] negatively impacted intention to stay [86]. Additionally, work-related injuries were reported at higher rates in older, female care workers [54], in migrant workers, and where assistive devices were not used for manual handling of residents [66,70], with an increased level of intention to leave [54].

The education and training of aged care staff were key determinants of retention. There were reports of insufficient onsite training before starting work [75,77,81,85], with some care workers feeling underprepared for both the physical demands of the role, and the emotional aspects of caring for older adults, especially at end of life [78,79,80]. Some migrant workers with overseas nursing qualifications reported the benefits of taking in-demand, lower-skilled roles in aged care whilst waiting to finalise nursing registration in their new country [56,72,88,93]. Others reported stress associated with a lack of career pathways, training, or formal education [51,73,82,86,88]. When career opportunities were available, they were often difficult to access financially or geographically, or not relevant to the scope of work [86]. Low local unemployment levels also impacted retention, with jobs in other industries, such as retail, offering similar salaries to aged care work, but with lower perceived work demands [58,77]. Facilitators to retention included promotion opportunities [69], schedule or roster management by the care workers, and safety training [90]. When a workforce was unionised with greater collective power [49] or a facility had lower proportions of residents with psychiatric illnesses [58], retention of care workers was higher.

The policy setting influenced worker behaviours. Policies regarding minimum ratios of professional to care worker staff or staff to resident ratios differed globally [60,62,68,69,70]. Supportive supervision levels were reported to be greater in the presence of higher professional-to-care worker ratios [60,70]. Whilst some studies reported the positive aspects of the requirement to have dedicated qualified nurses supervising care workers [60,68,87], there were significant challenges faced by nurses in managing a casualised, high-turnover workforce with need for continual training of new staff [89]. Despite efforts by governments worldwide to reform the working conditions of this sector, some care workers in our review expressed frustration with working in an unregulated wage system [86]. They felt that retention would improve if wage rises were associated with excellent performance [85] or length of tenure [89]. Increased rates of workplace injuries [70] and intention to leave [69] were reported when care worker staff-to-resident ratios were low. This was exacerbated by the COVID-19 pandemic, where restrictions on other health professionals placed an additional burden on care workers to perform duties beyond their scope of practice [83].

## 4. Discussion

Care staff are highly sought after worldwide yet retaining them in aged care roles is challenging. This review shows that the intention of personal care assistants, nurse assistants, and allied health assistants to stay working in the aged care sector was related to individual and organisational factors. Policies on wages, staff ratios and safety standards also impacted recruitment and retention. Our review identified studies from countries across European, Oceanian, Asian, and North American regions where life expectancy is typically longer [35]. We acknowledge that the nature of work for older adults’ care personnel in these regions will differ, as people spend more time in need of care, so knowledge of advance care planning and palliative care is required [95,96,97]. It is further noted that aside from population size and life expectancy, there are differences between countries regarding social security systems, impact of religion, and local economic situations [98,99]. These are some of the “key elements” of the older adult care workforce that affect turnover intention.

Our review highlighted that several individual factors were related to workforce retention. More positive lived experiences working in aged care were related to better retention, as was high job satisfaction. Retention was better when care workers felt supported by peers and supervisors and when there was capacity in their workloads to provide person-centred care. In agreement with a recent review on caring self-efficacy in direct care workers [100], we found that being able to establish compassionate relationships and meet the needs of residents was a key driver to remaining in the workforce [73,82,84]. Reducing sources of care worker stress also helped [73,75,84,85,86,89]. Evaluation of effectiveness of retention strategies was limited due to only three studies implementing an intervention.

Organisational factors also played a role. Retention was stronger when managers had positive leadership styles. Local procedures regarding staff rosters, shift lengths, split shifts [71,73,86], and enabling staff to contribute to roster management were facilitators [58,64,91]. The study by Brown et al. (2016) [50] described a staff responsibility approach where aged care workers were responsible for rostering. There was an enhanced ability to manage workloads, and this was associated with lower turnover rates compared to other care homes. As with recent government reports [36,101], we found that workplace health and safety was a predictor of retention, with workplace culture impacting whether occupational recommendations were adhered to [66,70,73,76]. For employers, organisational factors such as creating a positive workplace culture, ensuring good communication between leadership and staff, and ensuring a safe workplace through appropriate equipment- and facility-specific training aided care worker retention.

Across the world, there is a shift in aged care service provision to be more person-centred and value-based [18,102,103]. The quality of care remains dependent on recruiting and retaining very large numbers of the care workers providing the bulk of direct care to residents [104,105,106]. In recent years, care quality has come under increased scrutiny, and practice and policy concerns have been raised about staffing levels, recruitment, and retention [107]. Some have proposed that a need exists for better regulation of this workforce, with safety and quality of care at risk [36]. The recent Australian Royal Commission into Aged Care Quality and Safety (2021) [36] claimed that over 30% of people accessing residential or home-based aged care services experience substandard levels of routine personal or comprehensive care. Within our review, some care workers also shared the sentiment that care quality is not meeting basic expectations [84,85,89]. Some care staff expressed frustration with services not covering staff absences [77,78,79,84,89,90,91], insufficient pre-employment training [75,77,78,79,85], or poor cooperation across teams [69,75,77,90]. Some care workers felt underprepared for the emotional engagement required to deliver care in an empathetic and meaningful way. In some countries, the aged care workforce is multicultural and includes first- or second-generation migrants in low-paid roles [108]. Similar to other research in this area [109,110], migration of people suited for aged care work was hindered by visa pathways that channelled them into “low-skilled” nonprofessional care roles [72,81,93]. Migrant aged care workers are more likely to be on casual contracts [53,56]. Often, they seek more work hours, hold multiple jobs, and work at a lower skill level than afforded by their overseas qualifications [10,53,111]. A recurring theme was that the cultural diversity and cultural competence in the aged care sector needs to be optimised to accommodate care worker needs and to give staff opportunities for education and training [112].

There were several limitations of our review. Due to the low yield and heterogeneity of quantitative data, we were unable to complete a meta-analysis. Also, the review yielded few articles on retention of staff in home-based care, despite the preference of many older people to live in their own home as they age [113,114] and the known issues of staff shortages in this sector [110]. It was anticipated that findings would include examination of the impact of vaccine requirements during the COVID-19 pandemic, yet this did not emerge in the results. Also, we were not able to perform a detailed policy analysis due to variations across countries and exclusion of grey literature; this is recommended for future investigations.

## 5. Conclusions

Retention of aged care workers is a growing challenge worldwide. This systematic review summarised and aggregated contemporary evidence regarding retention of aged care workers, with analysis of retention strategy effectiveness limited by a low yield of interventional studies. This review highlighted the need for better support of care workers to keep them in employment. As well as optimising pay, workloads, and conditions, there is a need for reform of education and training, better career pathways, and more optimal support of worker wellbeing.

## Figures and Tables

**Figure 1 healthcare-11-03008-f001:**
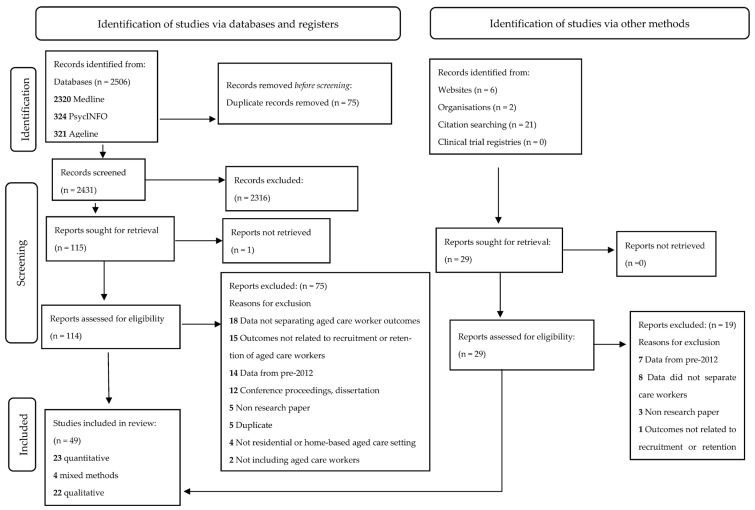
PRISMA flowchart [31] outlining study selection process.

**Table 1 healthcare-11-03008-t001:** Inclusion and exclusion criteria.

Inclusion	Exclusion
Aged care workers, such as nursing assistants, nurse aides, allied health assistants, personal care attendants, disability support workers employed in residential aged care facilities or home-based aged care. Primary qualitative, mixed-methods, or quantitative studies. Outcomes related to recruitment or retention of aged care support staff. Studies from 2012 to the present. Published in English.	Articles without data (i.e., editorials, commentaries). Conference proceedings, systematic, scoping or literature reviews, study protocols, policies, dissertations. Studies solely concerning psychometric properties of measurement tools. Studies related to registered health care professions, regulated workers or health professional students. Aged care volunteers or families providing unpaid care. Aged care employees not involved in direct care, e.g., gardeners, accountants, facility managers, case managers. Studies in acute care settings or specialised comprehensive cancer settings, hospices, or palliative care.

**Table 2 healthcare-11-03008-t002:** Overview of quantitative studies.

Author (Year) Country	Study Aims	Study Design, Data Collection	Participants	Key Findings	MMAT
Quantitative interventional studies					
Jeon (2015) [45] Australia	Evaluate the effectiveness of a clinical leadership training program (CLiAC) for middle managers in RACFs	RCT (double-blind cluster), questionnaires, human resource records	*n* = 589 IG: 191 AIN/PCAs, 26 nurses; CG: 276 AIN/PCAs, 46 nurses. 12 RACFs, 12 community	Positive influence on perceptions of supervisor support, management leadership styles, behaviours; passive avoidance behaviour of managers; overall satisfaction with leadership. No effect on care worker stress, turnover, intention to leave.	High
Ericson-Lindman (2017) [46] Sweden	Assess conscience, burnout, and social support of health care professionals following participatory action research	Pre–post intervention, questionnaires	*n* = 29 5 RNs and 24 CNAs from single RACF	Minor increase in perception of conscience and work-related social support. No effect on stress of conscience or burnout.	Mod
O’Brien (2019) [47] USA	Process evaluation of a group-based cognitive behavioural treatment program for care staff working in aged care	RCT with waitlist control, questionnaires	*n* = 71 Intervention 37, control group 34, nurses and NAs from NHs	Significant reduction in staff absences and mental health symptoms. No change in number of workplace injuries reported.	High
Quantitative descriptive studies					
Berridge (2020) [49] USA	Examine the relationship between NA retention and factors considered important to NH culture change at a facility level	Cross-sectional, questionnaire	*n* = 1368 NH administrators	Associations between high (vs low) retention rates of NAs and greater leadership/staff empowerment scores; low NH administrator turnover; high NH occupancy rates; presence of a union; greater hours per day allocated to residents.	High
Brown (2016) [50] USA	Identify differences in staffing hours, turnover, staff experiences, and perceptions between NHs and “Green House” NHs	Cross-sectional (two groups), questionnaires, human resource records	*n* = 502 (226 Green House, 276 NH), 47% CNAs	Green House NH CNAs reported increased ability to do their job in the event of staff absences. Nonsignificant trend towards lower CNA turnover in Green House NHs.	High
Chang (2021) [51] Taiwan	Investigate the influence of CNA job competency, satisfaction, and intention to stay	Cross-sectional, questionnaires	*n* = 333 236 CNAs from NHs, 97 CNAs from 26 NHs and 15 elderly welfare institutions	Job satisfaction related to higher job competency and salary, mediating a positive effect on intention to stay. Positively influenced by internal factors, i.e., self-esteem.	High
Chao (2020) [52] Taiwan	Assess determinants of intention to stay and actual retention rates between younger and older NAs	Longitudinal, questionnaires	*n* = 595 258 “Younger” and 337 “Older” CNAs from 137 LTCFs	Younger NA retention significantly influenced by gender and marital status. Older NAs retention significantly influenced by high work latitude and utilisation strategies; low burnout rates.	High
Charlesworth (2020) [53] Australia	Investigate the differences in migrant status amongst PCAs and HCWs against casual status and underemployment	Cross-sectional, questionnaires	*n* = 7114 2759 PCAs from RACFs, 4355 HCW	Migrant HCW and PCAs significantly more likely to be male and hold high levels of formal qualifications than locally born workers. Migrant workers much more likely to be underemployed and on casual contracts, with higher rates of multiple jobholding.	High
Cheung (2018) [54] Hong Kong	Determine prevalence of, and factors related to, workplace-related musculoskeletal injuries in Nas	Cross-sectional, questionnaires	*n* = 440 NAs from 47 NHs	Musculoskeletal injuries found to be experienced at high rates by NAs working in NHs; in mainly older, female workers who perceive work to be more stressful; associated with an increased intention to leave and a perceived health status of “not good”.	High
Dys (2022) [55] USA	Investigate staff perceptions of person-centred practices, organisational culture, and relationship to staff outcomes	Cross-sectional, questionnaire	*n* = 340 266 DCWs and 74 nurses from 23 NHs	No significant difference between DCWs or nurse perceptions of person-centred care; intention to leave; affective commitment.	High
Isherwood (2017) [56] Australia	To understand the difference between groups of Asian migrant care workers	Cross-sectional, questionnaire	*n* = 4530 PCAs from RACFs	Asian-born migrant PCAs had significantly higher levels of post-high school education compared to locally born PCAs. Asian PCAs were more likely to be employed on a casual basis, working fewer hours than desired, be expected to work longer shifts than scheduled and holding multiple jobs. Asian PCAs were more satisfied than locally born PCAs with the job role and wage.	High
Keisu (2018) [57] Sweden	Assess the relationship between care worker perceptions of management styles and ratings of effort and reward in aged care work	Cross-sectional, questionnaire	*n* = 159 80 nurses and allied health staff, 73 NAs, and 4 administrators from 9 RACFs	Association between professionals and greater reward for effort; professionals and identification of managers as having positive transformational leadership styles; higher transformational leadership style of managers and employee reward for effort.	High
Kennedy (2020) [58] USA	Examine facility-level factors associated with CNA turnover and retention in LTCF	Cross-sectional, questionnaire, certification reporting, human resources	*n* = 835 LTCFs	Associations between high turnover and lower retention of CNAs; for-profit status of facilities; turnover of DON staff; low empowerment of CNAs; greater proportion of residents with psychiatric illness (not dementia); low local unemployment rates.	High
Liang (2014) [59] Taiwan	Investigate the relationship between health-related quality of life and job stressors in NH care workers	Cross-sectional, questionnaire	*n* = 443 NAs from 64 LTCFs	Greater physical quality of life associated with staff with longer shift lengths; psychological job demands. Greater mental health quality of life associated with older staff age, working in smaller NHs.	High
Lin (2021) [60] China	Assess the level of supportive supervision and factors influencing this from RNs to PSWs in LTCFs	Cross-sectional, questionnaire	*n* = 643 PSWs from 12 LTCFs	Factors increasing supportive supervision included nurses with higher education levels; female nurses; completion of management training; increased years of experience; higher RN/PSW ratios. PSWs reported moderate levels of supportive supervision.	Mod
Matthews (2018) [61]	Examine the impact of manager-employee relationship quality on turnover in “low-wage earners” working in LTCFs	Longitudinal, questionnaire	*n* = 337 69 CNAs, 98 NAs, and 23 medical technicians	Turnover associated with reporting lower levels of affect, respect, or contribution towards supervisors; experience of lower levels of loyalty from supervisors than retained staff. Most significant factor irrespective of age, gender, or job satisfaction was loyalty experienced by the employee.	High
Rodríguez-Monforte (2020) [62] Canada	Analyse the association between work-related stress and job satisfaction of care staff in NHs and factors that may confound this relationship	Cross-sectional, questionnaire	*n* = 272 191 CNAs and 81 nurses from 5 NHs	Reduced job satisfaction related to stress resulting from resident behaviour; lower levels of supervisory support; less work effectiveness; less empowerment.	High
Rodwell (2015) [63] Australia	Examine the relationship between the job-demands resource model, negative affect, demographics workplace aggression	Cross-sectional, questionnaire	*n* = 291 208 nurses and 83 NAs from RACFs	CNAs reported high rates of threats of assault, physical assault, bullying, and emotional abuse from residents’ families. Increased emotional abuse was related to higher workloads, younger age of staff and lower levels of workplace support.	High
Sharma (2022) [48] USA	Assess the relationship between wages and nursing home staff turnover	Cross-sectional, human resource records	*n* = 439 NHs with turnover data from 2013 to 2017	Factors associated with lower turnover were higher wages, with small increases not effective. Factors not associated with lower turnover included resident demographics; staff hours; facility location; local unemployment rates.	High
Stevens (2022) [64] Denmark	Investigate the determinants of perceived quantitative work demands and at what organisational levels they occur in NHs	Cross-sectional survey and observational, questionnaire, observations, step rate	*n* = 383 185 CAs, 162 care helpers and 33 professional health care workers	Lower perceived quantitative work demands associated with least qualified staff (care helpers); migrants; working fixed night or evening/night shifts; lower work-related emotional demands; lower influence levels at work. Not associated with resident-related physical tasks or step counts.	High
Wallin (2012) [65] Sweden	Examine factors associated with job satisfaction in NAs working RACF	Cross-sectional, questionnaire	*n* = 225 Nas from specialised dementia care and general RACFs	Higher general job satisfaction associated with a caring climate and personalised care provision. Higher nursing care satisfaction associated with general work climate, environmental and organisational support. Low job satisfaction associated with NA health complaints.	High
Yang (2021) [66] Taiwan	Investigate the relationship between musculoskeletal disorders and psychosocial factors in NAs	Cross-sectional, questionnaire	*n* = 308 NAs from 15 LTCFs	Increased reporting of musculoskeletal injury related to lower social support; higher psychological job demands; older age of NAs; migrants; those not using assistive devices for manual handling tasks.	High
Zhang (2016) [67] USA	Explore psychological and organisational work-related factors that impact sleep in NAs working in LTCFs	Cross-sectional, questionnaires	*n* = 744 NAs from 15 LTCFs	Lower mental health scores related to work–family conflict; working night shifts; poor sleep quality, but not sleep quantity.	High

ACA: aged care assistant; CA: care aide; CG: control group; CNA: certified nursing assistant; DCW: direct care workers; DON: director of nursing; LN: licensed nurse; HCW: home care worker; HHA: home health aide; IG: intervention group; LTCF: long-term care facility; MMAT: mixed methods appraisal tool; NA: nurse assistant; NH: nursing home; NHA: nursing home administrator; PCW: personal care worker; PSW: personal support worker; RACF: residential aged care facility; RCT: randomised controlled trial; RN: registered nurse; SNF: skilled nursing facility.

**Table 3 healthcare-11-03008-t003:** Overview of mixed methods studies.

Author (Year) Country	Study Aims	Study Design, Data Collection	Participants	Key Findings	MMAT
Beynon (2022) [68] USA	Assess the perception (Aim 1) and explore the experience (Aim 2) of workplace collaboration amongst LNs and CNAs in RACFs	Cross-sectional, questionnaires, interviews	*n* = 116 (Aim 1). *n* = 36 (Aim 2) 68 CNAs and 48 LNs from 4 RACFs (Aim 1); 12 CNAs and 12 RNs from 4 RACFs (Aim 2)	Agreement between nurses and CNAs regarding moderate levels of collaboration and teamwork across facilities; belief that collaboration improved quality of care through a shared interest of resident wellbeing. LNs identified that due to hierarchy they were more likely to determine level of co-worker connection.	High
Dhakal (2020) [69] Australia	Explore factors related to attraction and retention of ACAs	Cross-sectional, questionnaire	*n* = 79 ACAs from 11 RACFs	Intention to leave associated with casual roles, younger age of workers, metropolitan location, heavy workloads, lack of teamwork, low staff-to-resident ratios. Retention associated with payrates; job security; promotional opportunities; available working hours.	High
Graham (2012) [70] USA	Investigate the prevalence of work-related back injuries in CNAs	Cross-sectional, questionnaire	*n* = 35 CNAs from NHs	Work related injuries were experienced by almost 50% of respondents. Majority back injuries incurred via patient manual handling. Low staff ratios and poor relationships with supervising nurses reported to be most difficult aspects of jobs.	Mod
Leskovic (2020) [71] Slovenia	Assess any change in job satisfaction and burnout of care worker staff in NHs from pre-pandemic to during a pandemic	Recurrent cross-sectional, questionnaire	*n* = 1188 1079 NAs, 109 nurses from 98 NHs	Burnout syndromes increased prevalence from 2013 to 2020; related to an increase in emotional exhaustion and lowered job satisfaction. Lower job satisfaction in 2020 associated with changes in shift length; lack of breaks worsened job satisfaction. Greater cooperation between staff was noted during the pandemic.	High

ACA: aged care assistant; CA: care aide; CG: control group; CNA: certified nursing assistant; DCW: direct care workers; DON: director of nursing; LN: licensed nurse; HHA: home health aide; IG: intervention group; LTCF: long-term care facility; MMAT: mixed methods appraisal tool; NA: nurse assistant; NH: nursing home; NHA: nursing home administrator; PCW: personal care worker; PSW: personal support worker; RACF: residential aged care facility; RCT: randomised controlled trial; RN: registered nurse; SNF: skilled nursing facility.

**Table 4 healthcare-11-03008-t004:** Overview of qualitative studies.

Author (Year) Country	Study Aims	Study Design, Data Collection	Participants	Key Findings	MMAT
Adebayo (2023) [72] Australia	Explore the perceptions of migrant care workers regarding job demands coping strategies and intention to stay	Interpretative descriptive, interviews	*n* = 20 Migrant care workers from RACFs	Motivators to working in aged care included employment availability, attraction to care role. Stressors included resettlement, communication difficulties with residents and co-workers, lack of familiarity with workplace routines, and experience of discrimination. Enablers of retention included flexibility and organisational support.	High
Amateau (2023) [73] USA	Explore trauma and resilience related concepts from CNAs experiences in RACF	Grounded theory, focus groups	*n* = 18 16 current CNAs and 2 former CNAs from 4 RACFs	Stress, strength, and resilience related to identity; relationships with residents; workplace culture and values; personal wellness.	High
Bergqvist (2022) [74] Sweden	Understand experiences of CNAs working in RACF during a pandemic	Interpretative descriptive, focus groups	*n* = 20 NAs from 4 RACFs	Major themes arising from work-related experiences included feelings of abandonment, disrespect, and fear; development of routines and strategies to cope; stress related to irregular staffing and management guidance.	High
Booi (2021) [75] Canada	Provide insight into the working conditions and perceptions of CAs working in a RACF	Long qualitative ethnographic study, observations, interviews	*n* = 31 CAs from 1 RACF	Main views of workers feeling unprepared, insufficient onsite training and stress related to staffing; powerlessness with relation to care decisions; a lack of respect and experience of stigmatisation; feeling overwhelmed and expressing an intention to leave, compelled to stay for resident wellbeing.	High
Cooke (2021) [76] Canada	Explore the impact of workplace incivility and bullying on residential CA relationships	Long qualitative ethnographic study, observations, interviews	*n* = 38 21 CAs, 6 LPNs, 5 administrative, and 6 support staff from 2 LTCFs	Exposure to incivility and bullying found to impact reluctance to request assistance or trust co-workers; sense of judgement around work ethic, with resentment towards staff receiving help from others; reinforcement of declining offers of assistance, leading to unsafe manual handling behaviours amongst newer staff.	High
Creapeau (2022) [77] USA	Explore SNF leadership and CNA staff perceptions of challenges related to CNA retention	Interpretative descriptive, interviews	*n* = 413 295 CNAs, 59 NH administrators, 59 DONs from 59 NHs	Key responses from stakeholder groups identified wages, shortage of candidates with appropriate qualifications, and nature of the job main causes of staffing challenges. NHAs also citing competition/low unemployment rates. All stakeholder groups agreed wages, working relationships, and appreciation the most important things for CNA retention.	Mod
Dijxhoorn (2023) [78] Netherlands	Understand NA experiences of providing end-of-life care in NHs	Interpretative descriptive, interviews	*n* = 17 NAs	Elements impacting care provision included suffering, grief, feelings of unfairness, accumulation of deaths; relationships and interactions with residents and families, receiving gratitude; feelings of fulfilment, powerlessness, and inadequacy.	High
Franzosa (2022) [79] USA	Understand the perspectives of staff working in NHs during a pandemic	Interpretative descriptive, interviews, focus groups	*n* = 62 56 CNAs and 6 administrators from 6 NHs	Identified challenges were staffing shortages, pressure to work when unwell or fatigued. Strategies to improve working conditions included teamwork and communication across direct care staff and management to ensure confidence and safety; accessibility of leadership, mentors, and ongoing training; recognition of work–life balance and mental health impact of working in pandemic conditions.	High
Gleason (2016) [80] USA	Understand the experience of HHAs following the death of a client	Interpretative descriptive, interviews	*n* = 80 HHAs working in community service for eldercare	Over 1/3 felt they could “very much” turn to supervisors for support; less than 1/5 felt they could turn to a co-worker. Low rates (less than 20%) sought support before or after client’s death from either supervisors or co-workers. Most common types of support sought were opportunity to talk; training related to death/dying; being allowed time off; consideration of notification methods to staff.	High
Goel (2015) [81] Australia	Understand the experience of migrant aged care workers in a regional area	Interpretative descriptive, focus groups	*n* = 7 5 PCAs, 1 HCW, 1 AHA	Flexibility of hours, care role, socialisation, and receiving wages improved satisfaction with employment. Time constraints, work demands, poor inter-collegial relationships, and lack of supervision and organisational support reduced satisfaction with employment.	Mod
Gray (2017) [82] USA	Explore CNA perceptions of work identity within context of relationships with other staff and residents in SNF	Interpretative descriptive, focus groups	*n* = 45 CNAs from 4 SNFs	Work-related identity categories were connector (communicating information about residents to supervisors). Provided CNAs with sense of control; advocate (CNAs felt to be in a strong position to detail any needs of residents); overloaded worker (impacted by limited or conflicting direction and emotional/physical exhaustion); companion (sense of importance to residents, compliments, and appreciation increased satisfaction).	High
Hoedl (2022) [83] Austria	Understand the experience of working during a pandemic for NH staff	Interpretative descriptive, interviews	*n* = 18 8 NA, 2 CA, 8 nurses from 5 NHs	Work-related consequences of the pandemic included increase in quantitative and qualitative workload—time resources for PPE, pressure to fill social needs of residents; changes to work organisation—positive communication from multidisciplinary team and organisation, negative aspect inability to take planned leave; physical—wearing PPE, tiredness, and exhaustion; psychological—uncertainty of the situation, stress, fear of infection for self and the residents; social—reduction in contacts outside of NH.	High
Holmberg (2013) [84] USA	Explore CNAs perceptions of work roles and work environment on care provision and mental health impacts	Interpretative descriptive, focus groups	*n* >150 CNAs from 7 NHs	Individual level themes were caregiver as both an identity and holistic practice, with experience of stress when quality care not achieved, concern about resident autonomy and dignity. Organisational level themes included issues with working for large companies, staffing shortages, and relationships with supervisors; perceived lack of respect from resident families or supervisors regarding CNA knowledge/skills or importance of their role.	High
Krein (2022) [85] USA	Understand the experience of NH staff and families of residents and perceptions of factors related to turnover	Interpretative descriptive, interviews	*n* = 42 16 CNA, 9 nurses, 17 family members of residents from 5 NHs	All stakeholder groups reported negative aspects including disruption to care, difference in quality of care, risk of errors. Minimal experience of turnover was reported within some family and NH administrators and noted positive aspects of high retention including consistency of care and increased teamwork. Proposed ways to reduce turnover included increases in wage potentially useful however, direct care staff and families felt this would need to be significantly high to make a difference; administrators and direct care staff identified rapport development, showing staff appreciation, and supporting teamwork.	High
Lim (2021) [86] Korea	Explore the influence of work-related care of the elderly on intention to stay and turnover in NH staff	Interpretative descriptive, interviews	*n* = 10 Care workers from 5 NHs and 5 home care providers	Factors influencing turnover intentions found to be low status of care work within society; employment instability, unprotected labour rights, and safety with absence of training and supervision; low wages and no regulation of wages; emotional/mental health stress.	High
Marziali (2015) [87] Canada	Explore the responses of NAs to a self-efficacy educational intervention	Interpretative descriptive, focus groups	*n* = 164 147 NAs and 17 nurses from 17 LTCFs	CNAs felt training provided novel ways to handle interactions with residents, but not necessarily reflective of complex work situations. Well-functioning teams reported CNA autonomy and feelings of respect for their role; poorly functioning teams reported more stress, little autonomy, and unsatisfactory relationships with supervisors.	High
Nichols (2015) [88] Australia	Investigate how multiculturalism shapes and is supported in aged care settings	Interpretative descriptive, interviews	*n* = 58 30 PCAs, 16 nurses, 5 management, and 5 family members from 6 RACFs	CaLD staff were more likely to have post-high school qualifications than non-CaLD with the majority obtained overseas. Opportunities for employment and aspirational lifestyle were drivers for migration. Migrant staff reported culture shock and limited understanding of dementia. CaLD staff reported experiences of discrimination and mistrust from non-CaLD colleagues.	High
Roussillon-Soyer (2021) [89] France	Investigate the psychological impact of absenteeism on nurse and NAs working in NHs	Interpretative descriptive, interviews	*n* = 42 11 nurses and 31 certified caregivers or noncertified caregivers from 7 NHs	Short-term or last-minute absences placed additional pressure on staff, reducing feelings of control, trust, and stability. Lack of training of recruits and substitutes increases safety risks to staff and residents. Lack of recognition of heavy workload, salaries not representative of contribution, reduction in quality of life due to often increased work hours. Work overload impacting satisfaction with job, pressured to neglect residents due to time constraints.	High
Senecal (2019) [90] USA	Understand factors influencing intention to stay in NAs working in SNFs	Interpretative descriptive, interviews	*n* = 10 NAs from 6 SNFs	Intention to stay supportive factors included work-related self-confidence; positive relationships with residents and appreciation; teamwork and consideration of others in own actions. Intention to stay threatening factors reported as seeking career advancement; difficulty with providing person-centred care due to resources; limited supervisory support and teamwork.	High
Sousa-Ribeiro (2022) [91] Sweden	Investigate the experience of older NAs employed in NH as they near retirement	Phenomenology, interviews	*n* = 8 NAs from 1 NH	Major themes included late-career plans with societal expected retirement age, openness to continue working with more flexibility in selected hours; personal health, work ability, and ageing; perception of work as a health risk due to physical and emotional stress; motives to continue working including financial considerations and meaningfulness of work.	High
Titley (2022) [92] Canada	Explore the impact on CAs working in LTCFs during a pandemic	Interpretative descriptive, interviews	*n* = 52 CAs from 8 LTCFs	Major themes arising included compounding stress associated with enforcing isolation; grief and loss—untimely death of residents; fear of infection for themselves and residents; significant staff shortages and limited direct communication with managers; increased feelings of resilience and optimism.	High
Winarnita (2022) [93] Australia	Understand the experience of Asian female migrant aged care workers in regional areas	Interpretative descriptive, interviews, observations	*n* = 7 PCAs from 7 RACFs	Major themes included overcoming prejudice from residents and locally born peers; the importance of peers with a similar cultural background; the benefits of working within an in-demand sector; the cost-effectiveness of working and living in a regional community; and the challenges of communication being an essential part of the role but coming from an English as a second language background.	High

ACA: aged care assistant; AHA: allied health assistant; CA: care aide; CaLD: culturally and linguistically diverse; CG: control group; CNA: certified nursing assistant; DCW: direct care workers; DON: director of nursing; LN: licensed nurse; HCW: home care worker; HHA: home health aide; IG: intervention group; LTCF: long-term care facility; MMAT: mixed methods appraisal tool; NA: nurse assistant; NH: nursing home; NHA: nursing home administrator; PCW: personal care worker; PSW: personal support worker; RACF: residential aged care facility; RCT: randomised controlled trial; RN: registered nurse; SNF: skilled nursing facility.

## Data Availability

No new data were created.

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
