# Peer review of "Factors Impacting Retention of Aged Care Workers: A Systematic Review"

_healthcare, 2023, doi:10.3390/healthcare11233008_

Round 1
Reviewer 1 Report
Comments and Suggestions for Authors
1. This study is a literature review aimed at identifying the retention factors among long-term care institutions and home care nursing staff. However, there seem to be unexpected keywords were listed. I recommend that the authors review the keywords, as there are many inappropriate or indefinite items. (for example, accidental falls, outcomes)
2. Within the category of long-term care institutions, there are various types such as adult day care or institutions with different natures. Can the authors clearly define and classify the content of relevant literature?
3. The 15th reference pertains to interventions on how nursing facilities can prevent falls among elderly residents. This content does not seem to be related to the focus of this study, let alone the incorrect citation of it on the second page, where it's incorrectly suggested that there is a connection between elderly falls and nursing staff retention.
4. In a systematic review study, citing the results of other systematic reviews might be somewhat inappropriate.
5. I recommend that you explicitly list the issues related to grey literature to inform what gaps in this topic still need to be addressed.
6. The issue of nursing staff shortages is well-established. From a practical standpoint, it might be more valuable to identify which strategies effectively increase retention. Evaluating the effectiveness of retention strategies would contribute more comprehensively to this topic.
Author Response
Reviewer 1
- This study is a literature review aimed at identifying the retention factors among long-term care institutions and home care nursing staff. However, there seem to be unexpected keywords were listed. I recommend that the authors review the keywords, as there are many inappropriate or indefinite items. (for example, accidental falls, outcomes). Thank you, we agree and have now edited this list (see line 40).
- Within the category of long-term care institutions, there are various types such as adult day care or institutions with different natures. Can the authors clearly define and classify the content of relevant literature?
Thank you, we agree that within long-term care facilities there are various groups, such as residential aged care, respite care, palliative care and adult day care. We have revised the manuscript in section 2.2 “eligibility and screening” line 118 to reflect your query with thanks and further note that a range of descriptors for care institutions is also detailed for each study in Supplementary Tables S4-S6.
- The 15th reference pertains to interventions on how nursing facilities can prevent falls among elderly residents. This content does not seem to be related to the focus of this study, let alone the incorrect citation of it on the second page, where it's incorrectly suggested that there is a connection between elderly falls and nursing staff retention. Thank you, we have amended this reference (see reference [15])to: Shin JH. Nursing Staff Characteristics on Resident Outcomes in Nursing Homes. J Nurs Res. 2019;27(1):1-9.
- In a systematic review study, citing the results of other systematic reviews might be somewhat inappropriate. Thank you, we have removed the citation of the systematic review and replaced it with a primary study (see reference [15]).
- I recommend that you explicitly list the issues related to grey literature to inform what gaps in this topic still need to be addressed.
We agree and have added new sentences on page 5, results section 3.1 lines 205-209 of the results “Grey literature was reviewed by two assessors. This identified gaps in research, including exploration of the specific needs of migrant care workers, the workforce challenges in respite aged care and strategies to improve recruitment of the aged care workforce, particularly in regional and remote communities.”
- The issue of nursing staff shortages is well-established. From a practical standpoint, it might be more valuable to identify which strategies effectively increase retention. Evaluating the effectiveness of retention strategies would contribute more comprehensively to this topic. We agree with this viewpoint and have added more text in results section 3.2, paragraph 1 and Discussion paragraph 2 to highlight that we only identified three studies that implemented strategies to improve retention.
“Three studies implemented strategies to improve retention. Of the studies applying an intervention, two were randomised controlled trials [1, 2] and one used participatory action research with pre-post assessments [3].”
“Reducing sources of care worker stress also helped [4-9]. Evaluation of effectiveness of retention strategies was limited due to only three studies implementing an intervention.”
Reviewer 2 Report
Comments and Suggestions for Authors
Title: Factors Impacting Retention of Aged Care Workers: A Systematic Review.
Reviewer Comments: Retention of care support workers in care facilities is a global challenge. These people have low job satisfaction rate and poorly defined career pathways. A mixed-methods systematic review of the workforce literature was conducted to understand the factors that attract and retain care staff across the aged care workforce. Authors identified 49 studies. Out of 49, three studies tested education and training interventions with the aim of boosting, workforce retention and the remaining 46 studies explored opinions and experiences of care workers. Authors identified a range of factors impacted retention of aged care workers. Authors found that Individual and organizational factors facilitating retention. Individual factors related to personal satisfaction with the job, good relationships with other coworkers, families, and residents. Organizational factors such as opportunities for career development, good salary, policies to prevent workplace injuries, and stable job.
Strengths:
1. Author did a good job in addressing the important factors impacting the retention.
2. Sample size and variations in the study group.
3. These kinds of studies will help governments to plan better for the benefit and retention of aged care workers.
Weaknesses:
1. There is no molecular data or experimentally derived data.
2. These types of studies are prone to bias, such as selection bias, attrition bias, and selective outcome reporting. Inconsistency that includes clinical or statistical heterogeneity. And imprecision that may lead to Type I and Type II errors.
3. Are these aged care workers are trained professionally? I mean professional skills including basic care knowledge pertaining to severe diseases. Or are they just good at the tasks of caring, cleaning and helping with hygiene.
4. Authors only focused on aged care workers. Are doctors, nurses and physician assistants and physiotherapists also included? If not including data from doctors or nurses also add lot of value to the work.
5. I feel that this study should have yielded more articles, but that’s not the case here.
6. Are these factors impacting retention is same for all countries or will change depending on the location?

Author Response
Reviewer 2
There is no molecular data or experimentally derived data. Given that this was a systematic review of previous studies already published in the literature, molecular data were not included. The Supplementary files provide detailed details on the data extracted for this review. The limitations of the type of data included reflect limitations in the published review and not the review process. This has been noted briefly in the limitations section (Discussion,line 196):
“There were several limitations of our review. Due to the low yield and heterogeneity of quantitative data, we were unable to complete a meta-analysis.”
These types of studies are prone to bias, such as selection bias, attrition bias, and selective outcome reporting. Inconsistency that includes clinical or statistical heterogeneity. And imprecision that may lead to Type I and Type II errors.
We agree with this statement, and our review now highlights the limited amount of research in this field with robust study designs (please see section 3.2 study characteristics, lines 212-216).
“Of the studies applying an intervention, two were randomised controlled trials [1, 2] and one used participatory action research with pre-post assessments [3]. The use of descriptive surveys and questionnaires with a cross-sectional (n=20) or longitudinal (n=2) design were reported in all quantitative descriptive and mixed-methods studies.”
Are these aged care workers are trained professionally? I mean professional skills including basic care knowledge pertaining to severe diseases. Or are they just good at the tasks of caring, cleaning and helping with hygiene.
The literature showed that the level of training varies significantly across the globe. We specified an aged care worker (Methods section 2.2) to be “a person employed to deliver direct care under the supervision of a registered or enrolled nurse [34] or under the supervision of an allied health professional or medical practitioner, within an aged care setting.” Requiring supervision to perform care tasks implies limited professional skills.
- Authors only focused on aged care workers. Are doctors, nurses and physician assistants and physiotherapists also included? If not including data from doctors or nurses also add lot of value to the work.
Our review did not include registered health professionals and the manuscript provides a justification for why “doctors and …” were not included in the current work on line 124-126 in methods section 2.2. “Analysis of the professional care workforce such as doctors, nurses and physiotherapists as this has been covered in a separate manuscript.”
- I feel that this study should have yielded more articles, but that’s not the case here.
We agree and believe this highlights the limited amount of research specific to this cohort.
- Are these factors impacting retention is same for all countries or will change depending on the location? This is a very good point. The factors impacting retention appear to differ across geographical regions, countries and cultures and this has been added in the Discussion paragraph 1 “Our review identified studies from countries across European, Oceania, Asian and North American regions where life expectancy is typically longer. We acknowledge that the nature of work for older adult’s care personnel in these regions will differ, as people spend more time in need of care, so knowledge of advance care planning and palliative care is required.”
Reviewer 3 Report
Comments and Suggestions for Authors
I read your article with great interest. The availability of human resources to care for the older adults is becoming an increasing challenge around the world. I believe that researching this is worthwhile. I would like to see the following revisions to improve the understanding of a wide range of readers.
The authors argue that the need for better support to keep care workers employed requires not only optimizing pay, workload, and working conditions, but also education and training reforms, better career paths, and more optimal support for worker well-being. This requires consideration of the "economic situation," "social security system," "religion," "population," "life expectancy," and other factors in each of the countries studied. These points were not readily apparent from the text.
Related to the above, we review reports from the United States, Australia, Canada, Sweden, China, Korea, and other countries. The World Health Organization's definition of the elderly is 65 years and older. However, these countries do not necessarily define 65 as elderly. This point needs to be clarified before the results can be discussed.
Are there differences in the nature of work for older adult’s care personnel in countries where life expectancy is longer than in countries where life expectancy is shorter? For example, in countries where life expectancy is longer, people spend more time in need of care, so knowledge of "advance care planning" and "palliative care" is required. These are the "key elements" of the older adult’s care workforce. Do these factors affect the "turnover intention" of the older adult’s care workforce? These points need to be clarified.
Without consideration of the above steps, it is difficult to determine the consistency of the authors' views.
That is all.
Please consider revising it.
Author Response
Reviewer 3
The authors argue that the need for better support to keep care workers employed requires not only optimizing pay, workload, and working conditions, but also education and training reforms, better career paths, and more optimal support for worker well-being. This requires consideration of the "economic situation," "social security system," "religion," "population," "life expectancy," and other factors in each of the countries studied. These points were not readily apparent from the text.
Thank you for this comment. We agree and have added another paragraph in the revised Discussion to explore this point: “It is further noted that aside from population size and life expectancy there are differences between countries regarding social security systems, impact of religion, and local economic situations. These are some of the “key elements” of the older adult’s care workforce that affect turnover intention.”
Related to the above, we review reports from the United States, Australia, Canada, Sweden, China, Korea, and other countries. The World Health Organization's definition of the elderly is 65 years and older. However, these countries do not necessarily define 65 as elderly. This point needs to be clarified before the results can be discussed.
We agree and have now noted this in methods section 2.2 lines 116-117. We chose to broaden our definition as “older adults” (see methods section 2.2, lines 119-120). We only included studies conducted in settings that referred to their service as delivering care to aged or elderly or geriatric populations.
Are there differences in the nature of work for older adult’s care personnel in countries where life expectancy is longer than in countries where life expectancy is shorter? For example, in countries where life expectancy is longer, people spend more time in need of care, so knowledge of "advance care planning" and "palliative care" is required. These are the "key elements" of the older adult’s care workforce. Do these factors affect the "turnover intention" of the older adult’s care workforce? These points need to be clarified.
Yes this is the case. We have added a new paragraph in the revised Discussion to say that “Our review identified studies from countries across European, Oceania, Asian and North American regions where life expectancy is typically longer. We acknowledge that the nature of work for older adult’s care personnel in these regions will differ, as people spend more time in need of care, so knowledge of “advance care planning” and “palliative care” is required. These are some of the “key elements” of the older adult’s care workforce that affect turnover intention.”
Reviewer 4 Report
Comments and Suggestions for Authors
Factors Impacting Retention of Aged Care Workers: A Systematic Review
The conciseness of the team writing is appreciated, but there are some pieces that need to be expanded on.
Eligibility and screening
The definitions need to be made clearer. In North America, which was the bulk of your study, there are private care homes where aged care workers may not be under the supervision of a professional. In Canada, specifically, we can have support workers that can be hired ‘off-the-street’ but take up to two years to receive any training – from a 10 week program to a 10 month program (unregulated). However, private care homes in Canada do not need to hire anyone with education in health at all. How did you determine that aged care workers were supervised by a health professional? OR, is there a role between home-based care and residential care that was not included? Do you have evidence to highlight for readers that there is another step in the trajectory of aged care homes that was not included or is a gap in your search?
The limitations
- Removes the reader from the importance of this review. Following a JBI methodology is very specific and rigorous. The limitations must be of the study as designed, not of what can be done by other work: 1) care staff was a choice (professional care workers is not a limitation), 2) vaccine requirements is a reach since it was mandated for many and the stigma of refusing would not be reported in this literature, but in COVID literature, 3) policy analysis would require inclusion of grey literature (the team’s choice to not include).
- Limitations for this study included not doing a meta-analysis for quantitative studies. While specific change may not be noted in so few studies, and while qualitizing quantitative data is expected in a review, this IS a limitation of this study.
- Another limitation is potentially including evidence that did not meet the highest criteria
The conclusions
- Two sentences is not enough for an overview of the paper and discussion results
- Significant to this work is the experience of migrant workers, but also workers who expressed concern over the standard of care
- The conclusions should address the aim of the study in lines 71-75
Author Response
Reviewer 4
The definitions need to be made clearer. In North America, which was the bulk of your study, there are private care homes where aged care workers may not be under the supervision of a professional. In Canada, specifically, we can have support workers that can be hired ‘off-the-street’ but take up to two years to receive any training – from a 10 week program to a 10 month program (unregulated). However, private care homes in Canada do not need to hire anyone with education in health at all. How did you determine that aged care workers were supervised by a health professional? OR, is there a role between home-based care and residential care that was not included? Do you have evidence to highlight for readers that there is another step in the trajectory of aged care homes that was not included or is a gap in your search?
Thank you for this insight. We agree that the definitions needed refinement and we have edited our definition in methods section 2.2 lines 113-114 to capture the scenario that you describe, where care workers are independently performing care tasks with overall supervision by a health professional (this could be a facility manager or care coordinator).
“An aged care worker was defined as a person employed to independently deliver direct care under the overall supervision of a registered or enrolled nurse [10] or under the supervision of an allied health professional or medical practitioner, within an aged care setting (residential aged care or home-based aged care.”
The limitations removes the reader from the importance of this review. Following a JBI methodology is very specific and rigorous. The limitations must be of the study as designed, not of what can be done by other work: 1) care staff was a choice (professional care workers is not a limitation), 2) vaccine requirements is a reach since it was mandated for many and the stigma of refusing would not be reported in this literature, but in COVID literature, 3) policy analysis would require inclusion of grey literature (the team’s choice to not include).
Thank you for your comments. Regarding 1) we have removed that sentence; 2) while we did not specifically search for “COVID literature”, several of our included studies used data gathered during the pandemic and referenced the effects of COVID on care worker retention, however we did not see analysis of vaccine refusal mentioned, hence our limitation comment; 3) thank you, we agree and have added this (final sentence in discussion).
“Also, we were not able to do a detailed policy analysis due to variations across countries and exclusion of grey literature; this is recommended for future investigations.”
Limitations for this study included not doing a meta-analysis for quantitative studies. While specific change may not be noted in so few studies, and while qualitizing quantitative data is expected in a review, this IS a limitation of this study. Another limitation is potentially including evidence that did not meet the highest criteria.
We have noted this as a new limitation in the discussion and revised the manuscript accordingly (discussion paragraph 6).
“There were several limitations of our review. Due to the low yield and heterogeneity of quantitative data, we were unable to complete a meta-analysis.”
The conclusions: Two sentences is not enough for an overview of the paper and discussion results. Thank you, we have extended the conclusions.
“Retention of aged care workers is a growing challenge worldwide. This systematic review summarised and aggregated contemporary evidence regarding retention of aged care workers, with analysis of retention strategy effectiveness limited by a low yield of interventional studies. This review highlighted the need for better support of care workers to keep them in employment. As well as optimising pay, workloads, and conditions, there is a need for reform of education and training, better career pathways and more optimal support of worker wellbeing.”
Significant to this work is the experience of migrant workers, but also workers who expressed concern over the standard of care.
Thank you for this comment, we have addressed this in the discussion of the revised manuscript “Within our review, some care workers also shared the sentiment that care quality is not meeting basic expectations.”
And “Similar to other research in this area, migration of people suited for aged care work was hindered by visa pathways that channelled them into ‘low-skilled’ non-professional care roles. Migrant aged care workers are more likely to be on casual contracts. Often, they seek more work hours, hold multiple jobs, and work at a lower skill level than afforded by their overseas qualifications. A recurring theme was that the cultural diversity and cultural competence in the aged care sector needs to be optimised to accommodate care worker needs and to give staff opportunities for education and training.”
The conclusions should address the aim of the study in lines 71-75. Thank you, we have extended the conclusions to address this point
Round 2
Reviewer 3 Report
Comments and Suggestions for Authors
Thank you for your careful revisions.
I have the impression that it is easier to read than last time.
I hope that it will be read by many readers and cited in the paper.